# MIXED PRECISION TRAINING OF CONVOLUTIONAL NEURAL NETWORKS USING INTEGER OPERATIONS

**Dipankar Das,**[*] **Naveen Mellempudi,**[*] **Dheevatsa Mudigere,**[*] **Dhiraj Kalamkar**[*]
{dipankar.das,naveen.k.mellempudi,
dheevatsa.mudigere,dhiraj.d.kalamkar}@intel.com
Parallel Computing Lab
Intel Labs, India

**Sasikanth Avancha, Kunal Banerjee, Srinivas Sridharan, Karthik Vaidyanathan, Bharat Kaul**
Parallel Computing Lab
Intel Labs, India

**Evangelos Georganas, Alexander Heinecke, Pradeep Dubey**       **Jesus Corbal**
Parallel Computing Lab                                           Product Architecture Group
Intel Labs, SC                                                   Intel, OR

**Nikita Shustrov, Roma Dubtsov, Evarist Fomenko, Vadim Pirogov**
Software Services Group
Intel, OR

## ABSTRACT

The state-of-the-art (SOTA) for mixed precision training is dominated by variants of low precision floating point operations, and in particular FP16 accumulating into FP32 Micikevicius et al. (2017). On the other hand, while a lot of research has also happened in the domain of low and mixed-precision Integer training, these works either present results for non-SOTA networks (for instance only AlexNet for ImageNet-1K), or relatively small datasets (like CIFAR-10). In this work, we train state-of-the-art visual understanding neural networks on ImageNet-1K dataset, with Integer operations on General Purpose (GP) hardware. In particular, we focus on Integer Fused-Multiply-and-Accumulate (FMA) operations which take two pairs of INT16 operands and accumulate results into an INT32 output.We propose a shared exponent representation of tensors, and develop a Dynamic Fixed Point (DFP) scheme suitable for common neural network operations. The nuances of developing an efficient integer convolution kernel is examined, including methods to handle overflow of the INT32 accumulator. We implement CNN training for ResNet-50, GoogLeNet-v1, VGG-16 and AlexNet; and these networks achieve or exceed SOTA accuracy within the same number of iterations as their FP32 counterparts without any change in hyper-parameters and with a 1.8X improvement in end-to-end training throughput. To the best of our knowledge these results represent the first INT16 training results on GP hardware for ImageNet-1K dataset using SOTA CNNs and achieve highest reported accuracy using half precision representation.

## 1 INTRODUCTION

While single precision floating point (FP32) representation has been the mainstay for deep learning training, half-precision and sub-half-precision arithmetic has recently captured interest of the academic and industrial research community. Primarily this interest stems from the ability to attain potentially upto 2X or more speedup of training as compared to FP32, when using half-precision

---
[*]Equal contribution

fused-multiply and accumulate operations. For instance NVIDIA Volta NVIDIA (2017) provides 8X more half-precision Flops as compared to FP32.

Unlike single precision floating point, which is a unanimous choice for *32b* training, half-precision training can either use half-precision floating point (FP16), or integers (INT16). These two options offer varying degrees of precision and range; with INT16 having higher precision but lower dynamic range as compared to FP16. This also leads to residues between half-precision representation and single precision to be fundamentally different – with integer representations contributing lower residual errors for larger (and possibly more important) elements of a tensor. Beyond this first order distinction in data types, there are multiple algorithmic and semantic differences (for example FP16 multiply-and-accumulate operation accumulating into FP32 results) for each of these data types. Hence, when discussing half-precision training, the whole gamut of tensor representation, semantics of multiply-and-accumulate operation, down-conversion scheme (if the accumulation is to a higher precision), scaling and normalization techniques, and overflow management methods must be considered in totality to achieve SOTA accuracy. Indeed, unless the right combination of the aforesaid vectors are selected, half precision training is likely to fail. Conversely, drawing conclusions on the efficacy of a method by not selecting all vectors properly can lead to inaccurate conclusions.

In this work we describe a mixed-precision training setup which uses:

- INT16 tensors with shared tensor-wide exponent, with a potential to extend to sub-tensor wide exponents.

- An instruction which multiplies two INT16 numbers and stores the output into a INT32 accumulator.

- A down-convert scheme based on the maximum value of the output tensor in the current iteration using multiple rounding methods like nearest, stochastic, and biased rounding.

- An overflow management scheme which accumulates partial INT32 results into FP32, along with trading off input precision with length of accumulate chain to gain performance.

The compute for neural network training is dominated by GEMM-like, convolution, or dot-product operations. These are amenable to speedup via specialized low-precision instructions for fused-multiply-and-accumulate (FMA), like AVX512_4VNNI [1]. However, this does not necessarily mean using half-precision representation for all tensors, or using only half-precision operations. In fact, performance speedups by migrating the compute intensive operations in both forward and back prorogation (FPROP, BPROP and WTGRAD) is often close to the maximum achievable speedup obtained by replacing all operations (for instance SGD) in half-precision. In cases where it is not, performance degradation typically happens due to limitations of memory bandwidth, and other architectural reasons.Hence on a balanced general purpose machine, a mixed-precision strategy of keeping precision critical operations (like SGD and some normalizations) in single precision and compute intensive operations in half precision can be employed. The proposed integer-16 based mixed-precision training follows this template.

Using the aforesaid method, we train multiple visual understanding CNNs and achieve Top-1 accuracies Russakovsky et al. (2015)on the ImageNet-1K dataset Deng et al. (2009) which match or exceed single precision results. These results are obtained **without changing any hyper-parameters, and in as many iterations as the baseline FP32 training**. We achieve 75.77% Top-1 accuracy for ResNet-50 which, to the best of our knowledge, significantly exceeds any result published for half-precision training, for example Micikevicius et al. (2017); Ginsburg et al. (2017). Further, we also demonstrate our methodology achieves state-of-the-art accuracy (comparable to FP32 baseline) with int16 training on GoogLeNet-v1, VGG-16 and AlexNet networks. To the best of our knowledge, these are first such results using int16 training.

The rest of the paper is organized as follows: Section 2 discusses the literature pertaining to various aspects of half-precision training. The dynamic fixed point format for representing half-precision tensors is described in Section 3. Dynamic fixed point kernels and neural network training operations are described in Section 4, and experimental results are presented in Section 5. Finally, we conclude this work in Section 6.

---

[1]https://www.anandtech.com/show/11741/hot-chips-intel-knights-mill-live-blog-445pm-pt-1145pm-utc

## 2 RELATED WORK

Using reduced precision for Deep learning has been an active topic of research. As a result there are a number of different reduced precision data representations, the more standard floating-point based Micikevicius et al. (2017); Ginsburg et al. (2017); Dettmers (2015) and custom fixed point schemes Vanhoucke et al. (2011); Courbariaux et al. (2014); Gupta et al. (2015); Hubara et al. (2016b); Köster et al. (2017).

The recently published mixed precision training work from Micikevicius et al. (2017) uses 16-bit floating point storage for activations, weights and gradients. The forward, back propagation computation uses FP16 computation with results accumulating into FP32 and a master-copy of the full precision (FP32) weights are retained for the update operation. They demonstrate a broad variety of deep learning training applications involving deep networks and larger data-sets (ILSVRC-class problems) with minimal loss compared to baseline FP32 results. Further, this shows that FP16/FP32 mixed precision requires loss scaling Ginsburg et al. (2017) to achieve near-SOTA accuracy. This ensures back-propagated gradient values are shifted into FP16 representable range and the small magnitude (negative exponent) values, which are critical for accuracy are captured. Such scaling is inherent with fixed point representations, making it more amenable and accurate for deep learning training.

Custom fixed point representations offer more flexibility - in terms of both increased precision and dynamic range. This allows for better mapping of the representation to the underlying application, thus making it more robust and accurate than floating-point based schemes.Vanhoucke et al. (2011) have shown that the dynamically scaled fixed point representation proposed by Williamson (1991) can be very effective for convolution neural networks - demonstrating upto to $4\times$ improvement over an aggressively tuned floating point implementation on general purpose CPU hardware. Gupta et al. (2015) have done a comprehensive study on the effect of low precision fixed point computation for deep learning and have successfully trained smaller networks using 16-bit fixed point on specialized hardware. With further reduced bit-widths, such fixed point data representations are more attractive - offering increased capacity for precision with larger mantissa bits and dynamically scaled shared exponents. There have been several publications with <16-bit precision and almost all of them use such custom fixed point schemes. Courbariaux et al. (2014) use a dynamical fixed point format (DFXP), with low precision multiplications with upto 12-bit operations. Building on this Courbariaux et al. (2015) proposed training with only binary weights while all other tensors and operations are in full precision. Hubara et al. (2016a) further extended this to use binary activations as well, but with gradients and weights still retained in full precision.Hubara et al. (2016b) proposed training with activations and weights quantized up to 6-bits and gradients in full precision. Rastegari et al. (2016) use binary representation for all components including gradients. However, all of the aforementioned use smaller benchmark model/data-sets and results in a non-trivial drop in accuracy with larger ImageNet data-set Deng et al. (2009) and classification task Russakovsky et al. (2015). Köster et al. (2017) have shown that a fixed point numerical format designed for deep neural networks (Flexpoint), out-performs FP16 and achieves numerical parity with FP32 across a wide set of applications. However, this is designed specifically for specialized hardware and the published results are with software emulation. Here we propose a more general dynamic fixed point representation and associated compute primitives, which can leverage general purpose hardware using the integer-compute pipeline. Further we provide actual accuracy and performance for training large networks for the ILSVRC classification task, measured on available hardware.

## 3 THE DYNAMIC FIXED POINT FORMAT

Dynamic Fixed Point (DFP) tensors are represented by a combination of an integer tensor $I$ and an exponent $E_s$, shared across all the integer elements. For the sake of convenience, the DFP tensor can be denoted as DFP-P = $\langle I, E_s \rangle$, where $P$ represents the number of bits used by the integer elements in $I$ (ex: DFP-16 contains 16-bit integers).

Figure 1 illustrates the differences in data representation between IEEE-754 standard format float, half-float and DFP-16 data format. DFP-16 data type offers a trade-off between float and half-float in terms of precision and dynamic range. When compared to full-precision floats, DFP-16 can achieve higher compute density and can carry higher effective precision compared to half-floats because of

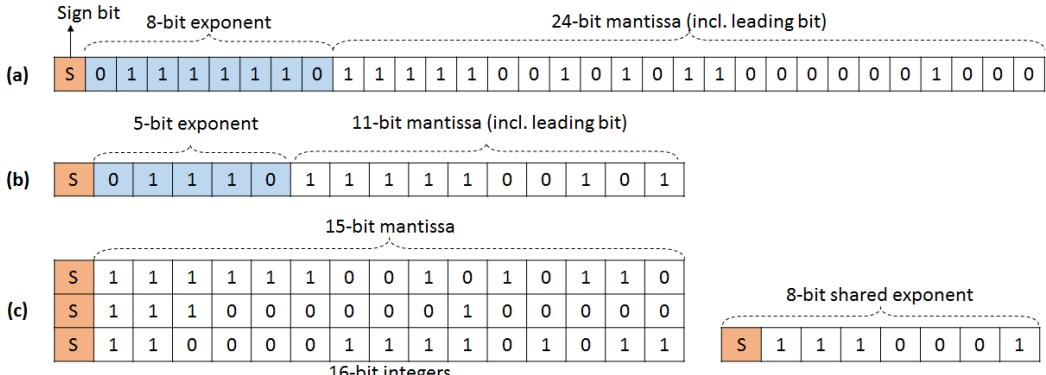

Figure 1: Snapshot of precision and dynamic range capabilities of a) IEEE-754 float b) IEEE-754 half-float, and c) Dynamic Fixed Point (DFP-16) data formats.

larger 15-bit mantissa (compared to 11-bits for half-floats). Further, the effective dynamic range of DFP format can be increased by extending the data type to use Blocked-DFP representation. Blocked-DFP uses fine-grained quantization to assign multiple exponents per tensor with smaller blocks of integers sharing a common exponent. Mellempudi et al. (2017) have demonstrated effectiveness of fine-grained quantization for low-precision inference tasks.

In this work, we use a single shared exponent for each tensor. The integers are stored in 2's complement representation and the shared exponent is an 8-bit signed integer. We use standard commodity integer hardware to perform arithmetic operations on DFP tensors. This implies that the exponent handling and precision management of DFP is done in the software, which is covered in more detail in Section 4.3.

### 3.1 DFP Tensor Primitives

To facilitate end-to-end mixed-precision training using DFP, we have created primitives to perform arithmetic operations on DFP tensors and data conversions between DFP and float. When converting floating point tensors into to DFP data type, the shared exponent is derived from the exponent of absolute maximum value of the floating point tensor. If $F$ is the floating point tensor, the exponent of the absolute maximum value is expressed as follows.

$$E_{fmax} = E\left(\max_{\forall f \in F} |f|\right) \tag{1}$$

The value of the shared exponent $E_s$ is a function of $E_{fmax}$ and the number of bits $P$ used by the output integer tensor $I$.

$$E_s = E_{fmax} - (P - 2) \tag{2}$$

The relationship of the resulting DFP tensor $\langle I, E_s \rangle$ with the input floating point tensor $F$ is expressed by Eq.3.

$$\forall i_n \in I, f_n = i_n \times 2^{E_s}, \text{where} f_n \in F \tag{3}$$

Extending this basic formulation Eq.3, we can define a set of common DFP primitives required for neural network training.

- Multiplying two DFP-16 tensors produces 32-bit $I$ tensor with a new shared exponent expressed as follows.

$$i_{ab} = i_a \times i_b \text{ and exponent, } E_s^{ab} = E_s^a + E_s^b \tag{4}$$

- Adding two DFP-16 tensors results in a 32-bit $I$ tensor and a new shared exponent.

$$i_{a+b} = \begin{cases} i_a + (i_b \text{>>}(E_s^a - E_s^b)), \text{ when } E_s^a \text{>} E_s^b \\ i_b + (i_a \text{>>}(E_s^b - E_s^a)), \text{ when } E_s^b \text{>} E_s^a \end{cases} \text{ and exponent, } E_s^{a+b} = \max_{E_s^a, E_s^b} \tag{5}$$

Note that when a Fused Multiply and Add operation is performed, all products have the same shared exponent: $E_s^{ab} = E_s^a + E_s^b$, and hence the sum of such products also has the same shared exponent.

- Down-Conversion scales DFP-32 output of a layer to DFP-16 to be passed as input to the next layer. The 32-bit $I$ tensor right-shifted $R_s$ bits to fit into 16-bit tensor. The $R_s$ value and the new shared exponent are expressed as follows.

$$R_s = A - (P + LZC(\max_{\forall i_{ab} \in I^{32}} |i_{ab}|))$$

$$i_{ab}^d = i_{ab} >> R_s \text{ and exponent, } E_s^{ab} + = R_s$$

(6)

In Eqn.6, A is accumulator bit-width, LZC( ) returns the leading zero bit-count.

## 4 NEURAL NETWORK TRAINING USING DYNAMIC FIXED POINT

Neural network training is an iterative process over mini-batches of data points, with four main operations on a given mini-batch: forward propagation (FPROP), back-propagation (BPROP), weight gradient computation (WTGRAD), and the solver (typically stochastic gradient descent, or ADAM).

In a CNN, the three steps of forward-propagation, back-propagation, and weight-gradient computation are often the compute intensive steps, and consist of GEMM-like (General Matrix Multiply) convolution operations which dominate the compute, and additional element-wise operations like normalization, non-linear (ReLU) and element-wise addition. In this work we propse a method to use INT16 operations, for implementing kernels for the convolutions and GEMM. There kernels are stitched with the rest of the operations in neural network training via *Dynamic Fixed Point* to floating point conversions described earlier in Section 3. In this section, we first describe the overall method for using dynamic fixed point in neural network training, and then explain the optimized kernel for convolutions.

### 4.1 TRAINING WITH DYNAMIC FIXED POINT

The mixed precision training scheme used in this work is described in Figure 2. The core compute kernels in this scheme are the *FP, BP, and WU* convolution functions which take two DFP-16 tensors as input and produces a FP32 tensor as output. For example *FP* accepts two DFP-16 tensors, $a_q$, *and* $w_q$ (activations and weights for layer-*l*), and produces a FP32 output tensorThe FP and BP operations are followed by quantization steps ($Q_a$, $Q_e$) which convert the FP32 tensors to DFP-16 tensors ($\hat{a}_q^l$, $e_q^l$) for operations in the next layer. The WU step is followed by the Stochastic Gradient Descent (SGD) step, which takes the FP32 tensor for weight-gradients ($\Delta w$) and a FP32 copy of the weights ($W^l$) as inputs, and produces an updated weight tensor as output. We follow the now established practice Micikevicius et al. (2017) of keeping a FP32 copy of weights as well as a low precision (DFP-16) copy of weights. Therefore SGD or other solvers are FP32 operations. In case a batch-norm layer is used, the DFP-16 tensors are loaded into registers and then the data is up-converted to FP32 to prevent overflows during stats computation.

### 4.2 CORE COMPUTE KERNELS

In this section we delve into efficient implementations of core compute kernels written using Integer FMA instruction sequence; in particular the AVX512_4VNNI instruction (described in Algorithm 1). This instruction takes a memory pointer as the first input and four vector registers as the second input and performs 8 multiply-add operations per output (16 Integer-OPs). For each 32b lane, the instruction takes two pairs of 16-bit Integers, performs a multiply followed by a horizontal add.

---

**Algorithm 1** Semantics of the QVNNI16 Instruction

---

```
1: K=4; SIMD_WIDTH=16;
2: QVNNI16(short *mem, _m512i vinp2[0..3], _m512i vout)
3: for v = 0 . . . K-1 do
4:     for o = 0 . . . SIMD_WIDTH-1 do
5:         vout[o] += vinp2[v][2*o]*mem[2*v] + vinp2[v][2*o+1]*mem[2*v+1]
6:     end for
7: end for
```

---

The *FPROP* convolution kernel is written using AVX512_4VNNI instruction in Algorithm 2. The data layout of the weights captures the 2-way horizontal accumulation operation in AVX512_4VNNI.

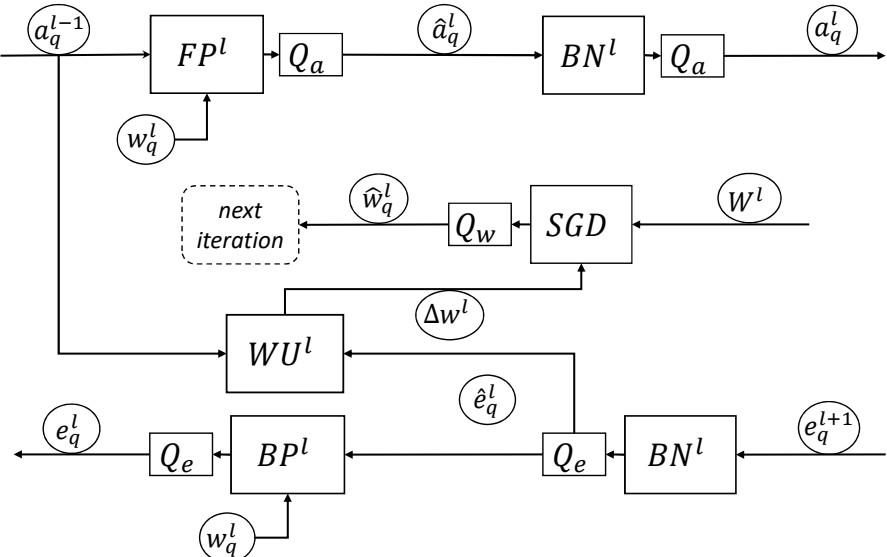

Figure 2: High-level data flow diagram for mixed precision training. Operators $FP^l$, $BP^l$ and $WU^l$ indicate convolution layers, while $Q_a$, $Q_w$, $Q_e$ are quantization operators for activations, weights and back propagated errors. Please note, the weight gradients ($\Delta w^l$) are not quantized before SGD, the updated weights are quantized for the next iteration.

Here the last dimension moves along consecutive input-feature maps. Hence the dimensions of activations is: N, C/16, H, W, 16, and that of weights is C/16, K/16, KH, KW, 8c, 16k, 2c (where C and K are input and output feature maps, H, W are input feature map height and width, and KH, KW are kernel height/width). Note that while we briefly touch upon data layout and blocking of the core kernel loops in Algorithm 2, detailed analysis of performance is not the objective of this work. These details are explored only to highlight different functional components of the kernel.

## 4.3 HANDLING OVERFLOWS IN INT16-INT32 FMAS

Multiplication of two INT16 numbers can result in a 30-bit outcome, and hence an accumulate chain of 3 products of INT16 multiplicative pairs can cause an overflow of the INT32 accumulator. In neural network training, accumulate chains can exceed a million in length (for example in the WTGRAD kernel).

One way to prevent overflows is to convert an INT32 intermediate output into FP32 before accumulation as described in lines 26-31 in Algorithm 2. Here we first convert the INT32 result to FP32 using the VCVTINTFP32 instruction, followed by a scale and accumulate into the final FP32 result using the VFP32MADD instruction. The scale used is $2^{(E_{inp}+E_{wt})}$ (equation 3), which is broadcast and stored in the *vscale* vector register. The instruction sequence in lines 26-31 can be applied after every AVX512_4VNNI instruction to prevent almost all overflows. However the overheads would be significant and hurt performance. Hence we pick the strategy of partial accumulations into INT32 for short accumulate chains, and subsequently converting the results into FP32.

*Performance Impact:* As outlined in Algorithm2 for performance we block additionally over the input feature maps (*ICBLK*) and use optimal register blocking (*RB_SIZE*). The difference between an ideal instruction sequence (with no overflow management) and Algorithm 2 is essentially the additional VCVTINTFP32 instruction (line 28). In the loop in lines 8-31, we have *(ICBLK/16)*KH*KW*2*RB* AVX512_4VNNI instructions, and *RB*4 + (ICBLK/16)*KH*KW*4* non-AVX512_4VNNI instructions, and *RB* VCVTINTFP32 instructions. The instruction overhead from overflow management therefore varies between <1% in most cases, to at most 3%.

The length of the accumulate chain (via sizing the input feature map blocking factor ICBLK in line 7) is selected to optimize instruction overheads and cache/instruction reuse. In this work we strive to keep the accumulate chain to more than 200 (which is empirically shown to be close to optimal). Often this accumulate chain also overflows, which we circumvent by shifting inputs. In this work, we shift both the inputs by 1-bit for all convolutions in all experiments. Hence effectively we have a DFP15 representation of all DFP tensors. It is notable that this shift value is largely dependent on this

---

**Algorithm 2** Example Forward Propagation Loop

---

1: fprop(DFP16 <input[IC/16][IH][IW][16], $e_{inp}$>, DFP16 <weights[IC/16][OC/16][KH][KW][8][16][2], $e_{wt}$>; FP32 output[OC/16][OH][OW][16] = 0)
2: _m512 vwt[0...3], vout[RB_SIZE], vtemp, vscale;
3: vscale = **VBROADCAST**($2^{(e\_inp+e\_wt)}$)
4: **for** ofm = 0 ... OC/16-1 **do**
5:  **for** ofh = 0 ... OH-1 **do**
6:   **for** ofw = 0 ... OW/RB_SIZE-1 **do**
7:    **for** ifm = 0 ... IC/ICBLK-1 **do**
8:     **for** rb=0 ... RB_SIZE-1 **do**
9:      vout[rb] = **SETZERO**()
10:     **end for**
11:     **for** ifmb = 0 ... ICBLK/16-1 **do**
12:      **for** kh = 0 ... KH-1 **do**
13:       **for** kw = 0 ... KW-1 **do**
14:        **for** ib = 0...1 **do**
15:         **for** v= 0...3 **do**
16:          vwt[v] = **LOAD**(&weights[ifm][ofm][kh][kw][ib*4+v][0][0])
17:         **end for**
18:         **for** rb = 0 ... RB_SIZE-1 **do**
19:          **AVX512_4VNNI**(&input[ifm*(IC/ICBLK)+icb][S*ofh+kh][S*ofw+kw][ib]), vwt[0...3], vout[rb])
20:         **end for**
21:        **end for**
22:       **end for**
23:      **end for**
24:     **end for**
25:    **end for**
26:    **for** rb=0 ... RB-1 **do**
27:     vtemp = **LOAD**(&output[ofm][ofmh][ofmw*RB_SIZE + rb][0])
28:     vout[rb] = **VCVTINTFP32**(vout[rb])
29:     vtemp = **VFP32MADD**(vtemp, vout[rb], vscale) //vtemp = vtemp + vout[rb]*vscale
30:     **STORE**(vtemp, &output[ofm][ofmh][ofmw*RB_SIZE + rb][0])
31:    **end for**
32:   **end for**
33:  **end for**
34: **end for**

---

inner accumulate chain length, and by constraining it we can find a shift value applicable across all operations. The combination of input shift and conversion of outputs to FP32 allows us to prevent occurrence of any overflows and hence catastrophic errors during training.

## 5 EXPERIMENTS AND RESULTS

We compare mixed precision DFP16 training with baseline full precision (FP32) for several ImageNet-class SOTA CNNs. Both baseline and DFP16 experiments are run using versions of the BVLC CAFFE framework Jia et al. (2014). For the baseline runs we use Intel's CAFFE branch[2]. For the mixed precision DFP16 experiments we use a private fork of this branch, where we have added DFP16 data-type support. The DFP16 compute primitives are supported through the prototype 16-bit integer kernels in Intel's MKL-DNN library[3] along with explicit exponent management as described in Section.4. Both baseline and mixed precision DFP16 experiments are run on the newly introduced Intel® XeonPhi™ Knights-Mill[4]. hardware using upto 32 nodes for training. Overall we see an average 1.8X speedup in the training throughput compared to the the baseline FP32 performance on the same platform.

---

[2] https://github.com/intel/caffe

[3] https://01.org/mkl-dnn

[4] Intel, Xeon, and Intel Xeon Phi are trademarks of Intel Corporation in the U.S. and/or other countries. Software and workloads used in performance tests may have been optimized for performance only on Intel microprocessors. Performance tests, such as SYSmark and MobileMark, are measured using specific computer systems, components, software, operations and functions. Any change to any of those factors may cause the results to vary. You should consult other information and performance tests to assist you in fully evaluating your contemplated purchases, including the performance of that product when combined with other products. For more information go to http://www.intel.com/performance

Table 1: Training configuration and ImageNet-1K classification accuracy

| Models | Batch-size / Epochs | Baseline | | Mixed precision DFP16 | |
|---|---|---|---|---|---|
| | | Top-1 | Top-5 | Top-1 | Top-5 |
| ResNet-50 | 1024 / 90 | 75.70% | 92.78% | 75.77% | 92.84% |
| GoogLeNet-v1 | 1024/ 80 | 69.26% | 89.31% | 69.34% | 89.31% |
| VGG-16 | 256 / 60 | 68.23% | 88.47% | 68.12% | 88.18% |
| AlexNet | 1024 / 88 | 57.43% | 80.65% | 56.94% | 80.06% |

## 5.1 ACCURACY RESULTS FOR CNNs

We trained several CNNs for the ImageNet-1K classification task using mixed precision DFP16: AlexNet Krizhevsky et al. (2012), VGG-16 Simonyan & Zisserman (2014), GoogLeNet-v1 Szegedy et al. (2015), ResNet-50 He et al. (2016). We use exactly the same batch-size and hyper-parameter configuration for both the baseline FP32 and DFP16 training runs (Table.1). In both cases, the models are trained from scratch using synchronous SGD on multiple nodes. In our experiments the first convolution layer (C1) and the fully connected layers are in FP32 (constituting about $5 - 10\%$ of compute for modern CNNs). Table.1 shows ImageNet-1K classification accuracies, training with DFP16 achieve SOTA accuracy for all four models and in several cases even better than the baseline full precision result.

To the best of our knowledge, *top-1 accuracy of **75.77%** and top-5 accuracy of **92.84%** for ResNet-50 with mixed precision DFP16 - is highest achieved accuracy on the ImageNet-1K classification task with any form of reduced precision training.*

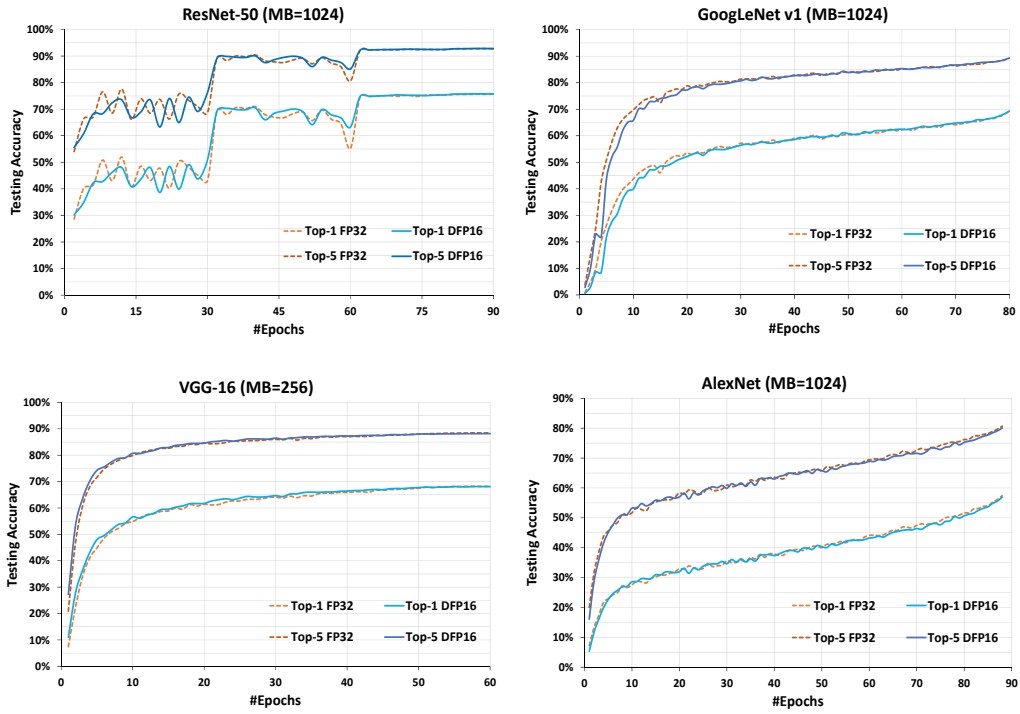

Figure 3: Convergence plots for DFP-16b training vs. reference baseline FP32 results for ResNet-50, GoogLeNet-v1, VGG-16 and AlexNet trained for ImageNet-1K classification task

It can be seen from Figure.3 that DFP16, closely tracks the full precision training. For some models like GoogLeNet-v1 and AlexNet, we observe the initially DFP16 training lags the baseline, however this gaps is closed with subsequent epochs especially after the learning rate changes. Further, we observe that compared to baseline run - with DFP16 the validation/test loss tracks much closer to the training loss. We believe this is the effect of the additional noise introduced from reduced precision computation/storage, which is results in better generalization with reduced training-testing gap and better accuracies.

## 5.2 PERFORMANCE DISCUSSION

For demonstrating the performance potential with mixed precision DFP16 training, we present detailed performance analysis and breakdown for the ResNet-50 topology as case study. The performance numbers reported below were measured on an Intel® XeonPhi™ Processor 7295 (codename Knights-Mill)[5] [4]

For the convolution kernels going from FP32 to DFP16, the $3 \times 3$ kernels are $1.8\times$ faster and the $1 \times 1$ kernels are $1.4\times$ faster; resulting in overall $1.5\times$ speedup. The baseline kernels include memory prefetch optimization, which when applied to DFP kernels should improve the performance by an additional $20\%$. The batchnorm computation is $2\times$ faster with DFP16, the speed up here is primarily due to $50\%$ bandwidth saving due to smaller memory footprint. In addition, the ReLU and EltWise layers are fused with batchnorm (Figure.4)) to avoid additional memory passes over the activation tensor. This fusion technique is orthogonal to mixed precision DFP16 training and can also be applied to baseline FP32 version as well, however its more relevant mixed precision DFP16 training due to faster compute. Furthermore, such memory bandwidth optimizations are becoming more critical with the growing disparity between compute capabilities and memory bandwidth with advent of specialized compute accelerators.

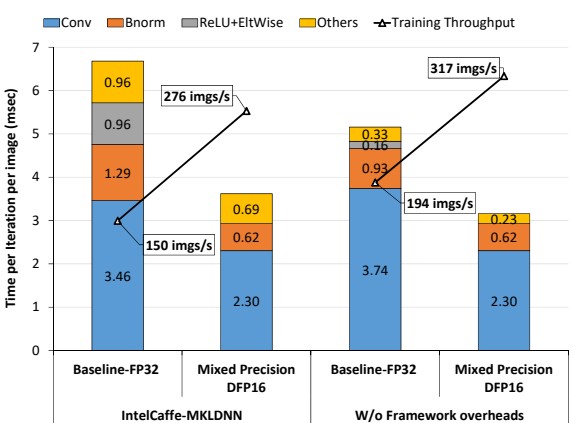

Figure 4: Performance breakdown of mixed precision DFP16 training vs. baseline FP32

With the above optimizations, we achieve an overall training throughput of 276 images/sec and 1.8X speed up over FP32 for ResNet-50. Additionally, we have improved SGD computation by $3\times$ over the standard implementation in Intel-Caffe, pushing the training throughput to 317 images/sec, shown as the framework overhead reduction in Figure.4. When exlpoiting similar tuning knobs, such as fusion and improved SGD, in case the of the baseline FP32 version its performance increases to 194 images/sec. Even in this case Mixed Precision DFP16 can yield a high speedup of 1.6X with respect to time-to-train.

## 6 CONCLUSIONS

We demonstrate industry-first reduced precision INT-based training result on large networks/data-sets. Showing on-par or better than FP32 baseline accuracies and potentially $2\times$ savings in computation, communication and storage. Further, we propose a general dynamic fixed point representation scheme, with associated compute primitives and algorithm for the shared exponent management. This DFP solution can be used with general purpose hardware, leveraging the integer compute pipeline. We demonstrate this with implementation of CNN training for ResNet-50, GoogLeNet-v1, VGG-16 and AlexNet; training these networks with mixed precision DFP16 for the ImageNet-1K classification task. While this work focuses on visual understanding CNNs, in future we plan to demonstrate the efficacy of this method for other types of networks like RNNs, LSTMs, GANs and extend this to wider set of applications.

ACKNOWLEDGMENTS

The authors would like to thank the Intel CRT-DC team that operates the Endeavor cluster and also the Excalibur cluster team for their outstanding support and assistance.

---

[5]https://ark.intel.com/products/128690/Intel-Xeon-Phi-Processor-7295-16GB-1_5-GHz-72-Core

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
