# OpenReview forum: "Mixed Precision Training of Convolutional Neural Networks using Integer Operations"
_ICLR.cc/2018/Conference — Accept (Poster)_

### Official Review · AnonReviewer3 · 2017-11-18
**SOTA with reduced precision on large CNNs**

**Rating:** 7
**Confidence:** 4

**Review:**

This paper describes an implementation of reduced precision deep learning using a 16 bit integer representation. This field has recently seen a lot of publications proposing various methods to reduce the precision of weights and activations. These schemes have generally achieved close-to-SOTA accuracy for small networks on datasets such as MNIST and CIFAR-10. However, for larger networks (ResNET, Vgg, etc) on large dataset such as ImageNET, a significant accuracy drop are reported. In this work, the authors show that a careful implementation of mixed-precision dynamic fixed point computation can achieve SOTA on 4 large networks on the ImageNET-1K datasets. Using a INT16 (as opposed to FP16) has the advantage of enabling the use of new SIMD mul-acc instructions such as QVNNI16.

The reported accuracy numbers show convincingly that INT16 weights and activations can be used without loss of accuracy in large CNNs. However, I was hoping to see a direct comparison between FP16 and INT16.

The paper is written clearly and the English is fine.

---

> ### Author Response · Authors · 2017-12-09
> **Performance Discussion**
>
> Thanks a lot for your comments.
>
> We do indeed have a Proof of Concept implementation for ResNet-50 on KNM 72c, 1.5GHz part with 16GB MCDRAM.
> On this part a FP32 implementation using MKLDNN on Intel Caffe achieves 152 img/s while our POC (also using Intel Caffe + MKLDNN interface (but not MKLDNN code)) achieved 275 img/s while achieving SOTA. This is a ~1.8x speedup over FP32. We also believe that there is scope for further improvements. If the PC/Reviewers permit we can add these results to the paper.
>
> Also the results in the paper are obtained using QVNNI-16 kernels on a 32 node KNM cluster as mentioned in Section 5.
>
> We do admit that the performance and overflow related discussion has room for improvement. Specifically the statement you point out pertains to the fact that we can always have the following sequence of instructions: QVNNI-16, cvtepi32ps (convert INT32 to FP32), fmaddps (scale and accumulate FP32 results) which will almost never overflow. Unfortunately the above mentioned sequence is ~3x slower than pure QVNNI-16 (as it has 3x more instructions). Therefore we select a compromise point between number of sufficient number of QVNNI-16 instructions followed by the convert and accumulate sequence, which optimizes performance without compromising numerics.
>
> I hope this clarifies things a little more. We will rewrite this Section 4.3 to clarify more.
>
> Again as per the breakup of performance lost per component of the mixed precision training methodology, if the reviewers/PC permits we can provide more details in the paper.
>
> We will update the submission shortly for a bunch of typographical and grammar issues we have identified at our end, and other edits discussed here.

---

### Official Review · AnonReviewer2 · 2017-11-22
**Mixed Precision Training**

**Rating:** 7
**Confidence:** 3

**Review:**

This paper is about low-precision training for ConvNets. It proposed a "dynamic fixed point" scheme that shares the exponent part for a tensor, and developed procedures to do NN computing with this format. The proposed method is shown to achieve matching performance against their FP32 counter-parts with the same number of training iterations on several state-of-the-art ConvNets architectures on Imagenet-1K. According to the paper, this is the first time such kind of performance are demonstrated for limited precision training.

Potential improvements:

  - Please define the terms like FPROP and WTGRAD at the first occurance.
  - For reference, please include wallclock time and actual overall memory consumption comparisons of the proposed methods and other methods as well as the baseline (default FP32 training).

---

> ### Author Response · Authors · 2017-12-12
> **Thank you**
>
> We would like to thank the reviewer for the comments.
>
> We will shortly update the manuscript to fix the missing definitions for the terms pointed out and also a number of other minor typographical errors that we have identified since submission.
>
> We intend to also include a more detailed discussion on performance (described in the comment below), in which we also include the baseline FP32 performance, along with a comparison with the INT16 variant in terms of various system aspects (memory footprint, performance profile...)

---

### Official Review · AnonReviewer1 · 2017-11-27
**New setup for CNN with half precision that gets 2X speedup on training**

**Rating:** 6
**Confidence:** 3

**Review:**

This work presents a CNN training setup that uses half precision implementation that can get 2X speedup for training. The work is clearly presented and the evaluations seem convincing. The presented implementations are competitive in terms of accuracy, when compared to the FP32 representation.  I'm not an expert in this area but the contribution seems relevant to me, and enough for being published.

---

> ### Author Response · Authors · 2017-12-12
> **Thank you**
>
> We would like to thank the reviewer for the comments.

---

### Author Response · Authors · 2018-01-06
**Updated revision of the paper**

We've added an updated revision of the paper which addresses the following :

> Typos and grammatic errors throughout the paper
> Added training throughput speedups (Section5)
> Included discussion on performance implications (Section4.3)

We thank all the reviewers for their helpful comments and feedback.

---

### Decision · Program_Chairs · 2018-01-29
**ICLR 2018 Conference Acceptance Decision**

**Decision:**

Accept (Poster)

**Comment:**

Mixed precision application of CNNs is being explored for e.g. hardware implementations of networks trained at full precision.  Mixed precision at training time is less common.  This submission primarily concerns itself with the practical implementation details of training with mixed precision, and focuses primarily on representation of mixed precision floating point and algorithmic issues for learning.  In the end the support for the approach is primarily empirical, with the mixed precision approach giving a factor of two speedup with half the precision, while accuracies remain effectively statistically tied on the ImageNet 1k database.  Table 1 should avoid the use of bold as there is likely no statistical significance.

The reviewers appreciated the paper. The proposed approach is sensible, and appears correct.